# Botulinum Toxin Type A (BoNT-A) Use for Post-Stroke Spasticity: A Multicenter Study Using Natural Language Processing and Machine Learning

**DOI:** 10.3390/toxins16080340

**Published:** 2024-08-02

**Authors:** María Jesús Antón, Montserrat Molina, José Gabriel Pérez, Santiago Pina, Noemí Tapiador, Beatriz De La Calle, Mónica Martínez, Paula Ortega, María Belén Ruspaggiari, Consuelo Tudela, Marta Conejo, Pedro Leno, Marta López, Carmen Marhuenda, Carlos Arias-Cabrales, Pascal Maisonobe, Alberto Herrera, Ernesto Candau

**Affiliations:** 1Department of Physical Medicine and Rehabilitation, Rio Hortega University Hospital, 47007 Valladolid, Spain; 2Department of Physical Medicine and Rehabilitation, University Hospital of Fuenlabrada, 28942 Madrid, Spain; 3Department of Physical Medicine and Rehabilitation, Son Espases University Hospital, 07210 Palma de Mallorca, Spain; 4Department of Physical Medicine and Rehabilitation, General University Hospital, 12004 Castellón, Spain; 5Department of Physical Medicine and Rehabilitation, University Hospital Puerta de Hierro-Majadahonda, 28222 Madrid, Spain; 6Savana Research S.L, 28004 Madrid, Spain; carias@savanamed.com; 7Department of Biometry, Ipsen Pharma, 92100 Boulogne-Billancourt, France; 8Medical Affairs, Ipsen Pharma, 28050 Madrid, Spain

**Keywords:** ischemic stroke, post-stroke spasticity, botulinum toxin A, electronic health records, natural language processing, artificial intelligence, machine learning

## Abstract

We conducted a multicenter and retrospective study to describe the use of botulinum toxin type A (BoNT-A) to treat post-stroke spasticity (PSS). Data were extracted from free-text in electronic health records (EHRs) in five Spanish hospitals. We included adults diagnosed with PSS between January 2015 and December 2019, stratified into BoNT-A-treated and untreated groups. We used EHRead^®^ technology, which incorporates natural language processing and machine learning, as well as SNOMED CT terminology. We analyzed demographic data, stroke characteristics, BoNT-A use patterns, and other treatments. We reviewed the EHRs of 1,233,929 patients and identified 2190 people with PSS with a median age of 69 years; in total, 52.1% were men, 70.7% had cardiovascular risk factors, and 63.2% had suffered an ischemic stroke. Among the PSS patients, 25.5% received BoNT-A at least once. The median time from stroke to spasticity onset was 205 days, and the time from stroke to the first BoNT-A injection was 364 days. The primary goal of BoNT-A treatment was pain control. Among the study cohort, rehabilitation was the most common non-pharmacological treatment (95.5%). Only 3.3% had recorded monitoring scales. In conclusion, a quarter of patients with PSS received BoNT-A mainly for pain relief, typically one year after the stroke. Early treatment, disease monitoring, and better data documentation in EHRs are crucial to improve PSS patients’ care.

## 1. Introduction

Stroke is the second leading cause of disability and death worldwide [1], affecting approximately 13.7 million people annually [2], including 1.12 million people in the European Union [3], with a prevalence of 187 cases per 100,000 person-years in Spain [4]. Post-stroke spasticity (PSS) is a common complication that affects nearly one-third of stroke patients and often develops within 1–4 months after the stroke, causing a noticeable decrease in the patients’ quality of life [5,6,7]. PSS is a motor and sensory disorder causing increased involuntary tonic muscle stretch reflexes, which leads to the shortening of muscles and soft tissues [8]. Spasticity is typically more frequently experienced in the upper extremities and is often accompanied by pain and disability [5,9]. Treatment for PSS often involves an interdisciplinary approach, including both pharmacological and non-pharmacological interventions such as long-duration stretches (limb casting or splinting), exercise, oral muscle relaxants, focal treatments to improve the physical function of limbs [6], and extracorporeal shock wave treatment [10,11]. Early and effective treatment is critical to prevent symptoms such as muscle contractures, stiffness, and pain [12].

Local intramuscular injection with botulinum neurotoxin type A (BoNT-A) is an established and well-tolerated first-line pharmacological treatment to manage focal spasticity [13]. There are three currently approved preparations of BoNT-A in Spain—abobotulinumtoxinA (aboBoNT-A), incobotulinumtoxinA (incoBoNT-A), and onabotulinumtoxinA (onaBoNT-A). Each of these preparations presents different pharmacological features, which could be responsible for the observed variations in clinical response, including differences in dosing, duration of action [13], and immunogenicity [14]. Moreover, other factors can affect a patient’s response such as individual anatomy, dose–response relationship, treatment reconstitution, and length of storage after reconstitution [15]. These challenges, along with the diversity of symptoms, treatment goals, and variability in the use of BoNT-A, in terms of doses administered, treatment intervals, and concomitant treatments, among others [15,16], make it difficult to draw robust conclusions about the management of PSS in clinical practice. Despite these discrepancies, increasing evidence shows that early BoNT-A treatment (four to six weeks after a stroke) is key for improving PSS symptoms [17,18,19]. However, there is a lack of consistency in national data registries across the globe that makes it difficult to calculate the prevalence and management of PSS, emphasizing the room for improvement in prevention strategies and clinical stroke care [20,21]. Therefore, there is a need to better understand the real-world characteristics of PSS patients and the use and treatment goals of BoNT-A in this patient population to guide future research and improve patients’ outcomes.

The information in patients’ electronic health records (EHRs) represents an important source of real-world data (RWD), avoiding the selection bias that clinical trials usually present by requiring strict inclusion and exclusion criteria. Moreover, they contain unstructured clinical notes, which better describe patients’ clinical characteristics, management, and treatment journeys within hospital settings compared to structured information [22,23]. Natural language processing (NLP) and machine learning (ML) are areas within artificial intelligence that are able to analyze and contextualize written and oral texts and have been recently employed to extract free-text information from EHRs [24,25]. In this regard, they provide great potential to help clinicians extract valuable data from patients’ EHRs and aid researchers in creating cohorts from real-world scenarios. This approach enhances the understanding of diseases by providing a more comprehensive and realistic view of clinical characteristics and treatment patterns compared to studies relying exclusively on structured data (such as International Classification of Diseases or ICD) or clinical trials [26,27,28,29]. Additionally, it facilitates the development of predictive tools for various conditions, including stroke [25].

This study aimed to describe the demographic and clinical characteristics, as well as the treatment patterns of patients with PSS in a real-world setting in Spain using NLP and ML focusing on the use of BoNT-A to identify potential areas for improvement and to optimize treatment in PSS patients. This approach will allow us to better understand this patient population and its complexity, as well as the management of PSS in actual clinical scenarios.

## 2. Results

### 2.1. Study Population and Stroke Characteristics

After analyzing the EHRs of 1,233,929 patients, we identified and included in the study 2190 individuals with PSS, of whom 559 (25.5%) received at least one BoNT-A treatment, while 1631 (74.5%) received none. Figure 1 shows the distribution of the population included in the study. Out of the total patients treated with BoNT-A, 204 (36.5%) had no information regarding the type of preparation they received.

Table 1 shows the main patient and stroke characteristics along with all included patients and the two subgroups, BoNT-A-treated and non-BoNT-A-treated.

More than half (*n* = 1140; 52.1%) of the overall population was male, with fewer males in the BoNT-A-treated group than in the untreated group (43.8% and 54.9%). The median (Q1, Q3) age of the patients was 69 (57, 79) years. Patients treated with BoNT-A were slightly younger than those who were not treated [64 (53, 74) and 71 (59, 81) years, respectively]. At least one risk factor for cardiovascular disease (CVD) was recorded in 1549 (70.7%) patients of the overall sample, with a higher frequency in the non-BoNT-A-treated than in the treated group (76.1% and 54.9%). Hypertension was the most frequent CVD risk factor, occurring in 1319 (85.2%) patients, followed by dyslipidemia (*n* = 918, 59.3%) and diabetes mellitus (*n* = 611, 39.4%). Other comorbidities were reported in 995 (45.4%) patients, among which the most frequently observed were valvopathies (*n* = 546, 54.9%) and atrial fibrillation (*n* = 438, 44.0%). Most comorbidities were numerically more common in the non-BoNT-A-treated than in the BoNT-A-treated group, such as hypertension (86.6% and 79.5%), dyslipidemia (59.7% and 57.7%), atrial fibrillation (46.6% and 32.0%), and atherosclerosis (10.0% and 6.2%).

Ischemic was the most common stroke type found in the overall population, as well as in the BoNT-A-treated and non-BoNT-A-treated groups (63.2%, 60.0%, and 64.2%, respectively). The most frequently reported stroke location was the middle cerebral artery, which occurred in 420 (72.9%) cases. Regarding stroke sequelae, hemiparesis (45.0%) and hemiplegia (44.5%) were the most frequent ones. The median (Q1, Q3) time from stroke to the first mention of spasticity was 205 (32, 615) days. This time was longer in the subset of patients treated with BoNT-A [344 (121, 835) days] than in those not treated [173 (23, 544) days] (Table 1).

### 2.2. Spasticity-Affected Areas and Muscular Groups

We recorded information about spasticity areas in 391 (70.0%) patients from the BoNT-A-treated group and in 1092 (67.0%) patients from the non-BoNT-A-treated group. In both study subgroups, both the upper limb (UL) and lower limb (LL) were the most affected areas (43.7% and 40.3% of cases, respectively). Within the BoNT-A-treated group, more patients experienced spasticity in the UL compared to the LL (35.5% and 20.7%). In contrast, the non-BoNT-A-treated group exhibited more spasticity in the LL (40.1%) compared to the UL (19.5%). The muscular groups most affected by spasticity were the muscles responsible for elbow flexion in the UL (*n* = 552, 48.8%), while the muscular groups associated with equinovarus foot pattern predominated in the LL (*n* = 331, 34.3%). Table 2 shows the spasticity-affected areas and muscular groups in all patients, as well as in both subgroups of patients.

### 2.3. BoNT-A Treatment

Among patients treated with BoNT-A, the median (Q1, Q3) time between stroke and the first mention of BoNT-A in the EHRs was 364 (152, 850) days. onaBoNT-A was the most common preparation for 272 (48.7%) treated patients, followed by aboBoNT-A for 70 (12.5%) patients, and incoBoNT-A for 13 (2.3%) patients.

Regarding the use of BoNT-A preparations according to PSS affected area, onaBoNT-A was also the most used preparation in patients with UL spasticity, LL spasticity, or both (87.9%, 72.9%, and 58.9%, respectively). The use of aboBoNT-A was more frequent in patients with LL spasticity and in those with spasticity in both the UL and LL. Specifically, 22.9% of patients with LL spasticity were treated with aboBoNT-A, and 33.3% of patients with spasticity affecting both limb sets. In contrast, only 12.0% of patients with UL spasticity received aboBoNT-A. Moreover, incoBoNT-A was the least used preparation found in 7.7% of patients with both UL and LL spasticity, in 4.2% of patients from the LL spasticity group, and was not reported in the UL spasticity group (Figure 2).

Among those 355 patients with recorded information about the specific BoNT-A preparation (onaBoNT-A, aboBoNT-A, or incoBoNT-A), only 24 (6.7%) patients switched treatment to a different BoNT-A preparation during the study period. The median (Q1, Q3) time between BoNT-A injections was 135 (105, 170) days. At least one treatment goal was found in 489 patients (87.5%) from the BoNT-A-treated group, in which pain relief was the most frequently observed (*n* = 485,99.2%) (Figure 3).

### 2.4. Other Treatments

Among PSS patients, non-pharmacological treatment was more common than pharmacological treatment (51.2% and 36.4%, respectively). The most common non-pharmacological treatment was rehabilitation and physical therapy in the overall PSS, BoNT-A-treated, and non-BoNT-A-treated groups (95.5%, 91.6%, and 96.9%, respectively). Diazepam was the most frequently used pharmacological treatment among all groups (45.4% in overall PSS, 47.2% in the BoNT-A-treated group, and 44.7% in the non-BoNT-A-treated group) (Table 3).

### 2.5. Spasticity Monitoring Scales

Spasticity assessment tools were only reported in 73 (3.3%) patients. The classical and modified Ashworth scale (any scale) were the most frequently detected (*n* = 60; 82.0%). The Tardieu and Visual Analog scales (any scale) were reported in 11 (15.0%) and 2 (3.0%) patients, respectively. The use of goniometry or goal attainment scales was not reported in the EHRs.

### 2.6. EHRead^®^ Performance Evaluation

EHRead^®^ performance evaluation showed a strong capability to detect critical variables defining the study population and those related to BoNT-A administration (Table 4).

## 3. Discussion

In this multicenter observational study, using advanced NLP and ML techniques, we conducted a detailed extraction and subsequent analysis of secondary data from the EHRs of patients with PSS in Spain. Through this analytical approach, we identified and described a cohort of 2190 patients with PSS and stratified them in two subgroups based on the presence or absence of BoNT-A treatment. The analysis revealed that PSS patients were predominantly elderly adults, approximately 70 years of age, and had several CVD risk factors and comorbidities. In this regard, nearly three-quarters of patients presented at least one risk factor for CVD, with hypertension and dyslipidemia being the most frequent risk factors across all groups. In addition, valvulopathies and atrial fibrillation were also frequently detected, with both being well-documented risk factors for stroke [2]. The presence of paresis, which is a major predictive factor of spasticity, was documented at a higher frequency than expected [26]. Moreover, most of these individuals had experienced an ischemic stroke within the middle cerebral artery, which is largely consistent with previous studies [2,27,28,29].

Our results show that PSS patients treated with BoNT-A were younger and had a higher female ratio than PSS untreated patients. The lower prevalence of BoNT-A use in older individuals may arise from various factors, including the propensity for younger patients to receive more proactive interventions, the adequacy of the therapeutic efforts among the elderly, increased mortality, or heightened disability from frequent medical incidents or comorbidities [30]. On the other hand, the increased usage in women could be related to the fact that females tend to suffer from more severe post-stroke complications than males [31,32]. We reported a median of 364 days between stroke and the first BoNT-A injection, which is longer than a previous study reported in French patients who were treated for spasticity within 285 days [33]. Despite the absence of a clear consensus about the optimal time for stroke patients to receive their first BoNT-A treatment for PSS, current recommendations indicate that treatment should be initiated as early as possible to maximize its effectiveness and improve the patient’s function and quality of life [8]. While current evidence suggests that PSS develops within 1–3 months after stroke, our results may reflect a late use of this treatment in our setting. However, these data may be influenced by multiple factors, such as prior treatment outside the hospital setting, which was not be reported in the EHRs analyzed.

In the overall study population, PSS occurred about seven months after the stroke, while in those receiving BoNT-A treatment, it was reported one year post-stroke. However, there was a significant temporal variability for the inoculation, ranging from as early as one month to as late as two years post-stroke. New evidence suggests that BoNT-A treatment within three months post-stroke reduces spasticity [18], suggesting that early treatment is key in preventing long-term PSS health issues. Although no large-scale studies have examined the natural history of spasticity and contracture development, it has been reported that the incidence of joint range loss can increase with time, ranging from 27% at one month to 43% at six months [34]. It is worth mentioning that the later development of PSS reported in the EHRs in our study could be related to the absence of in-hospital reports when spasticity is diagnosed in post-stroke patients. In this sense, a lot of these patients are usually referred to external rehabilitation units after the initial stroke episode, and it is plausible that PSS appeared during this period of convalescence. Therefore, the first mentions of PSS, as well as the first BoNT-A treatment, in the EHRs of our study should not be taken as the PSS diagnosis date, which could have been determined previously in other specialized centers.

The most common areas affected by spasticity in PSS patients were both the UL and LL across all groups (overall, BoNT-A-treated, and untreated patients). Elbow flexion was the most commonly affected area in the UL, and the equinovarus foot was the most common in the LL, which is in agreement with other studies [5,6].

We found that about a quarter of patients with PSS were treated with BoNT-A. Recent studies in European and Asian countries have concluded that BoNT-A treatment can improve patient outcomes such as pain relief, muscle tone, and improved motor control [35,36,37,38]. However, several studies have evaluated the use of BoNT-A treatment in real-world scenarios, also showing low rates of BoNT-A treatment. Levy J et al. found that 21.5% of PSS patients received at least one injection of BoNT-A based on an analysis of data extracted from the French National Hospital Discharge Database [33]. Conversely, another study that analyzed the sales database information from the Swedish healthcare system revealed that 9.2% of adult patients with disabling spasticity received BoNT-A within a one year period [39]. The authors of these studies highlighted the BoNT-A treatment underuse and underscored the need for a consensus about clinical practice. They suggested that the potential reasons may include a limited awareness among physicians about clinical practice guidelines (despite the recommendation of BoNT-A as a first-line pharmacological treatment for spasticity), coupled with a lack of access to specialists that are capable of administering BoNT-A injections [13].

The most common preparation used was onaBoNT-A, followed by aboBoNT-A and incoBoNT-A. Based on the current available evidence, no data have demonstrated the superiority of one formulation over another in terms of efficacy, safety, or the area affected by PSS [6,13,34]. Head-to-head studies are ongoing [40], which may shed light on the possible differences observed in duration between marketed products [41]. Our results show that aboBoNT-A was the most administered BoNT-A treatment in patients with LL spasticity or both LL and UL involvement. Conversely, a recent study in Asian patients with PSS revealed that 94.1% received aboBoNT-A injections in the UL, resulting in the successful management of most spasticity symptoms [35]. Another recently published real-world, retrospective study in patients with PSS from the United States concluded that onaBoNT-A was the most commonly used formulation to treat UL spasticity [14]. However, our methodology does not allow us to infer causality between the prescribed treatment and potential reasons for the specific choice reflected in the free-text of EHRs, so we cannot determine whether the use of one formulation over another is motivated by effectiveness or safety data, or simply by availability or the treating physician’s experience with one of the available formulations.

Pain relief was the treatment goal for nearly all patients who received BoNT-A treatment, which is in agreement with other studies that have reported this symptom to be one of the top goals among PSS patients who receive BoNT-A treatment [13,42]. Approximately half of the total included patients received concomitant non-pharmacological treatment, with the most common being rehabilitation and physiotherapy, following expert consensus recommendations [5,15]. The same consensus reported that oral anti-spasticity drugs have not proven to be as effective and are more associated with systemic side effects compared to focal BoNT-A treatment [5].

Scales such as Ashworth, Tardieu, Visual Analog Scale, goniometry, and goal attainment are supposedly used in routine clinical practice since they are designed to help identify early risk factors for PSS [5,9,43,44]. However, these scales were not widely reflected in the EHRs of included patients in our real-world setting. The absence of reported clinically relevant variables, such as these clinical scales, poses a challenge in determining whether the missing data are due to clinicians not reporting it in the EHRs or because baseline visits containing this information could have been conducted in many cases outside the hospital environment, such as in rehabilitation centers where patients could have been referred after the stroke event. However, scarce monitoring scale registration is an interesting finding previously described in other RWD studies [45]. Importantly, not finding these reported indices does not necessarily mean that healthcare providers do not use them, but that they are not recording them. This gap in clinical documentation may undermine the ability for accurate disease monitoring, and outcome assessments point out the need for an improvement in medical care.

The main strength of this study is the novel technology used, which allows us to interpret RWD in EHRs and gain novel insight into this understudied patient population and their treatment patterns. The use of unstructured information from EHRs through NLP has been found to have a much higher sensitivity than structured queries [46]. Moreover, unlike previous epidemiological studies conducted in Spain using ICD [47], our use of refined SNOMED clinical terms can be used directly for healthcare provider input and is, therefore, more appropriate for capturing RWD [48,49]. Finally, our novel NLP technology enabled the capture of EHR data across multiple centers nationwide, which can be further aggregated and analyzed to answer important clinical questions [47].

Despite these advantages, we also recognize some limitations. First, given that this study relies on free-text RWD, the potential number of variables included in the analyses was limited by the information contained in the EHRs. Regarding that, it was seen a lack of reporting on clinically relevant variables, such as spasticity location, dosages, and clinical scores. In addition, PSS per se has been shown to be under-reported in EHRs possibly due to a lack of consensus on the diagnosis, heterogeneity in methods, time frame of assessing spasticity, and poor previous published data. Then, in this cohort, some patients with PSS could have been not included if not previously recognized [20]. Then, proper data entry in patients’ EHRs and international reporting standards are necessary to improve data quality when a secondary use is performed for research, as well as to calculate disease risk, indexes or scores, and guide clinicians toward optimal treatments [50]. Importantly, the use of not only unstructured data, but also structured data such as pharmacy or laboratory data, should be considered to improve these results. Second, we did not have data from different centers where patients would have been treated after the stroke such as rehabilitation hospitals. This impedes the identification of a variable as a “true zero”, i.e., missing data not reported by clinicians in the EHRs or missing data that were never entered into the EHRs due to previous treatment in a non-hospital clinical setting. Moreover, in this context, the first mentions of PSS or BoNT-A in the analyzed EHRs may not be related to diagnosis or first treatment dates. Third, since our technology is based on EHRs where the sequence of events may not always be confirmed, we cannot infer causality between the detected treatment goals and the BoNT-A treatment. Finally, due to the descriptive nature of this study, outcomes related to the effectiveness or safety of BoNT-A were not evaluated, so further studies in real-world settings should be performed to evaluate them in PSS patients following BoNT-A treatment.

## 4. Conclusions

This study represents the largest cohort of patients with PSS in Spain to date. NLP and ML techniques allowed us to provide a comprehensive description of the clinical characteristics, spasticity involvement, and treatments used by these patients, with a special focus on the use of BoNT-A. Through our analyses of patient EHRs, we discovered that patients with PSS had a very complex profile with a high burden of comorbidities, likely reflecting the need for multidisciplinary management. Additionally, only one-quarter of PSS patients were treated with BoNT-A, primarily for pain relief, with a mean time of one year between the stroke event and the first administration. This suggests a delay in the diagnosis and treatment of these complications, highlighting room for improvement in the early detection and treatment of spasticity in patients who have suffered from a stroke. The successful application of NLP techniques to access and analyze EHRs depends on a multidisciplinary effort to improve how clinicians document their routine practice in patients’ records. This study provides valuable information for better understanding and properly managing this post-stroke complication.

## 5. Materials and Methods

### 5.1. Study Design and Study Population

This was a multicenter, retrospective, and observational study using secondary data captured in the EHRs of adult patients (aged ≥ 18 years) with a history of stroke and the presence of PSS or BoNT-A treatment described during the study period (1 January 2015 to 31 December 2019). Patients with PSS were stratified depending on whether they received BoNT-A treatment during the study period (BoNT-A and non-BoNT-A groups) or not. A retrospective cross-sectional analysis was conducted at the index date defined as the earliest date when either “spasticity” or “BoNT-A treatment” terms were found in the EHR. Treatment-related variables and PSS monitoring scales were analyzed during the follow-up period, ranging from index date to the latest EHR within the study period. (Figure 4).

This study was conducted in five hospitals located in four different regions within the Spanish National Healthcare Network—Madrid (Hospital Universitario de Fuenlabrada, Hospital Universitario Puerta de Hierro-Majadahonda), Balearic Islands (Hospital Universitari Son Espases), Castile and Leon (Hospital General Universitario Río Hortega), and Valencia (Hospital General Universitari de Castelló).

### 5.2. Data Source and Extraction

The unstructured free-text information in EHRs was collected from all available records and departments in the participating hospitals (including inpatient hospitals, outpatient hospitals, and emergency rooms). Unstructured clinical data were extracted and analyzed using the EHRead^®^ technology following previously described methods [22,23]. Briefly, the free-text information from de-identified EHRs was extracted and organized using the SNOMED CT terminology encompassing codes, synonyms, and definitions from clinical documentation. This data-driven technology relies on NLP and ML to generate a synthetic anonymized database that contains any detection of medical concepts and associated metadata in the source population. EHRead^®^ performance was externally validated, as previously described [51]. This evaluation consisted of a comparison between EHRead^®^ reading output and an annotated corpus of the same EHRs by expert physicians in each participating site of the study (standard to compare). The level of agreement between EHRead^®^ output and the standard was expressed in terms of precision (positive predictive value), recall (sensitivity), and their harmonic mean F1-score, which balances precision and recall in a single metric. Additional details regarding EHRead^®^ technology, as well as its performance evaluation, are provided in the Appendix A section.

### 5.3. Study Variables

The study variables were extracted and analyzed as part of a curation process that guaranteed the quality and integrity of the data. This process involved medical experts in NLP and a committee of 15 physicians specialized in physical medicine and rehabilitation with extensive experience in the field, who elaborated and curated the full list of specific study variables.

General patient characteristics (demographics and comorbidities), stroke-related data (etiology, vascular territory, sequelae, and time from stroke to spasticity), spasticity information (affected areas such as UL, LL, or both areas, and the muscular groups affected per area), spasticity assessment scales, BoNT-A treatment (commercial preparation, changes between preparations, time from stroke to BoNT-A treatment, and treatment goals), and other treatments for spasticity were extracted from the EHRs at index date or during the follow-up. To reconstruct patient history, all information coming from the same participating center before index (including information stemming from EHRs dated before the study period start, if available) was analyzed. For variables analyzed around discrete time points, the closest value to the time point (within reference time windows) was taken. Reference time windows accounted for the variability in healthcare management between patients, specialists, and hospitals, maximizing data retrieval from EHRs. The time window ranges for each variable or group of variables are detailed in table footnotes.

### 5.4. Data Analysis

In our descriptive analysis, categorical variables were presented as frequencies to illustrate the distribution of different categories within the dataset. Numerical variables were summarized using medians and quartiles (Q1, Q3) to convey central tendency and variability without being influenced by outliers. Percentages were calculated based on the number of non-missing observations, ensuring accurate reflection of the available data. Missing data were handled according to the nature of the data collection process and assuming that physicians reflect clinically relevant information in the EHRs. In this context, the absence of a particular term referring to a specific comorbidity was treated in the same way as if it was a negated comorbidity (i.e., the patient has no hypertension). This approach ensured a consistent and clinically meaningful analysis. Data analysis was performed using “R” software (version 4.0.2) and Python (version 3.7.12).

## Figures and Tables

**Figure 1 toxins-16-00340-f001:**
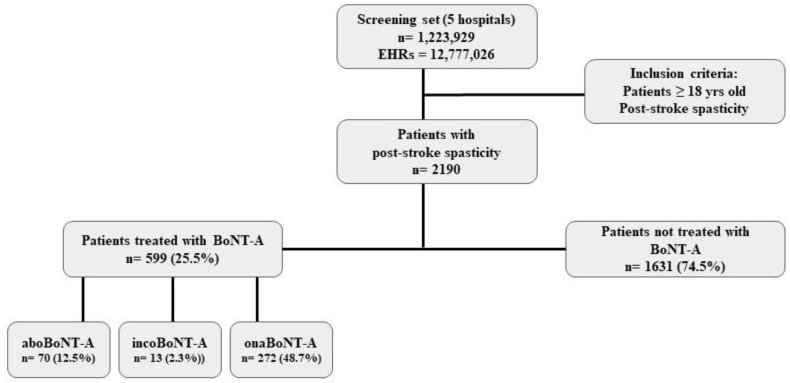
Flow chart of PSS patients’ subgroups: patients treated with a BoNT-A preparation and patients not treated with a BoNT-A preparation. aboBoNT-A: abobotulinumtoxinA; incoBoNT-A: incobotulinumtoxinA; onaBoNT-A: onabotulinumtoxinA.

**Figure 2 toxins-16-00340-f002:**
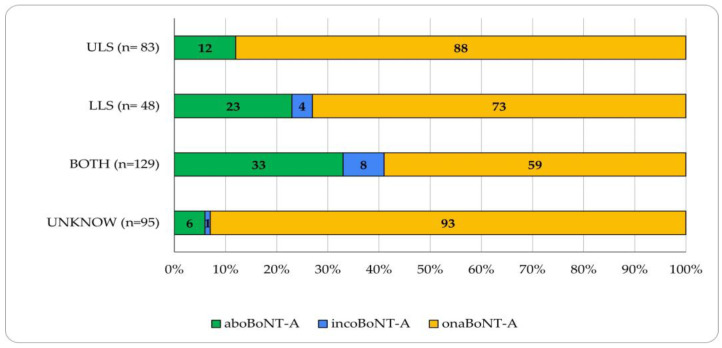
Types of BoNT-A treatment according to spasticity pattern. Graphic representation of patients who received BoNT-A treatment according to spasticity pattern—upper limb spasticity (ULS), lower limb spasticity (LLS), and both (ULS and LLS). The bars are divided into colors depending on the type of BoNT-A treatment received. aboBoNT-A: abobotulinumtoxinA; incoBoNT-A: incobotulinumtoxinA; onaBoNT-A: onabotulinumtoxinA. Out of the 355 patients with recorded information about the specific BoNT-A formulation received, 260 patients had data on both the specific BoNT-A formulation and spasticity pattern, while 95 had data on the specific formulation, but the spasticity patterns were unknown. All are included in this figure.

**Figure 3 toxins-16-00340-f003:**
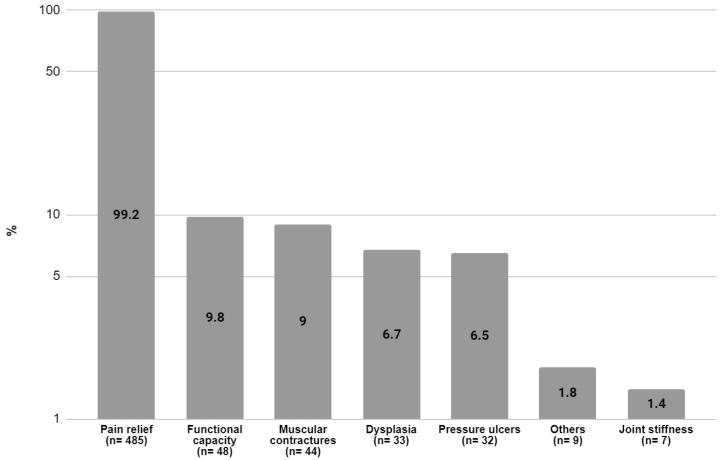
Patient treatment goals. Data from 489 (87.5%) patients treated with BoNT-A where at least one treatment goal was detected. “Others” refers to other treatment goals that included impaired movement, improvement in quality of life, and persistent abnormal posture. Please note that in some cases, patients had more than one feature, so the sum of patients might add up to more than 100%.

**Figure 4 toxins-16-00340-f004:**
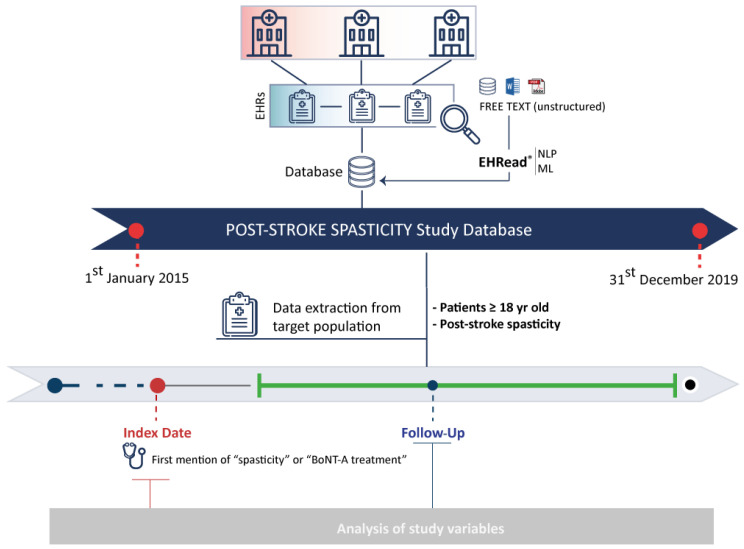
Study design. Baseline data were extracted using different time windows around the respective Index Date. The follow-up data were extracted from index date to the latest data point available.

**Table 1 toxins-16-00340-t001:** PSS patients characteristics at baseline.

	Overall PSS (*n* = 2190)	BoNT-A (*n* = 559)	non-BoNT-A (*n* = 1631)
General Characteristics			
Demographics			
Gender, male	1140 (52.1)	245 (43.8)	895 (54.9)
Age, years	69 (57, 79)	64 (53, 74)	71 (59, 81)
CVD risk factors	1549 (70.7)	307 (54.9)	1242 (76.1)
Hypertension	1319 (85.2)	244 (79.5)	1075 (86.6)
Dyslipidemia	918 (59.3)	177 (57.7)	741 (59.7)
Diabetes mellitus	611 (39.4)	106 (34.5)	505 (40.7)
Obesity	263 (17.0)	58 (18.9)	205 (16.5)
Comorbidities	995 (45.4)	178 (31.8)	817 (50.1)
Valvulopathies	546 (54.9)	101 (56.7)	445 (54.4)
Atrial fibrillation	438 (44.0)	57 (32.0)	381 (46.6)
Obstructive sleep apnea	147 (14.8)	37 (20.8)	110 (13.5)
Transient ischemic attack	132 (13.3)	25 (14.0)	107 (13.1)
Atherosclerosis	93 (9.3)	11 (6.2)	82 (10.0)
Stroke characteristics *			
Stroke etiology ^&^	1826 (83.4)	435 (77.8)	1391 (85.2)
Ischemic	1154 (63.2)	261 (60.0)	893 (64.2)
Hemorrhagic	672 (36.8)	174 (40.0)	498 (35.8)
Vascular territory ^&^	576 (26.3)	110 (19.6)	466 (28.5)
Middle cerebral artery	420 (72.9)	84 (76.4)	336 (72.1)
Internal carotid artery	102 (17.7)	16 (14.5)	86 (18.5)
Posterior circulation	34 (5.9)	7 (6.4)	27 (5.8)
Anterior cerebral artery	20 (3.5)	3 (2.7)	17 (3.6)
Stroke sequelae ^&^	1331 (60.7)	280 (50.0)	1051 (64.4)
Hemiparesis	598 (45.0)	130 (46.4)	468 (44.5)
Hemiplegia	593 (44.5)	120 (42.9)	473 (45.0)
Others	302 (22.7)	60 (21.4)	242 (23.0)
Time from stroke to spasticity, days ^#^	205 (32, 615)	344 (121, 835)	173 (23, 544)

BoNT-A: botulinum neurotoxin type A. CVD: cardiovascular disease. The denominator of percentages of the subcategories in bold is based on the N of the overall sample—patients treated with BoNT-A and patients not treated with BoNT-A. The denominator of percentages in indented rows is based on the N of the parent category. Categorical variables are expressed as frequencies n (%), and numerical data are expressed as median (Q1, Q3). Data were extracted and analyzed considering all available information from the first hospital report to one month post index date. * In some cases where categories are non-exclusive, patients have more than one feature, so the sum of patients might add up to more than 100%. When two exclusive categories were detected, rules were used to assign the broadest (extension) or the most severe (plegia) categories. No cases of triparesis or triplegia were detected. ^&^ The data reflect the number (*n*) and percentage (%) of patients with available information for this variable. ^#^ Time from stroke to spasticity onset was calculated for 957 patients (196 BoNT-A, 761 non-BoNT-A), including only those with stroke before the first spasticity mention.

**Table 2 toxins-16-00340-t002:** Spasticity areas and muscular groups affected.

	Overall PSS (*n* = 2190)	BoNT-A (*n* = 559)	Non-BoNT-A (*n* = 1631)
Spasticity affected area, *n* (%) ^&^	1483 (67.7)	391 (70.0)	1092 (67.0)
Upper and lower limb affected	612 (41.3)	171 (43.7)	441 (40.3)
Lower limb affected	519 (35.0)	81 (20.7)	438 (40.1)
Upper limb affected	352 (23.7)	139 (35.5)	213 (19.5)
Muscular groups affected per area			
Upper limb muscular groups, *n* (%) *^&^	1131 (51.6)	310 (55.5)	821 (50.3)
Elbow flexion	552 (48.8)	175 (56.5)	377 (45.9)
Others	241 (21.3)	112 (36.1)	129 (15.7)
Shoulder adduction	119 (10.5)	75 (24.2)	44 (5.4)
Thumb in palm	72 (6.4)	55 (17.7)	17 (2.1)
Claw hand muscles	20 (1.8)	20 (6.5)	0 (0.0)
Wrist flexion	13 (1.1)	13 (4.2)	0 (0.0)
Lower limb muscular groups, *n* (%) *^&^	964 (44.0)	252 (45.1)	712 (43.7)
Equinovarus foot	331 (34.3)	13 (5.2)	200 (28.1)
Hip flexion	300 (31.1)	93 (36.9)	207 (29.1)
Knee extension	119 (12.3)	40 (15.9)	79 (11.1)
Others	11 (1.1)	1 (0.4)	10 (1.4)

BoNT-A: botulinum neurotoxin type A. Data were extracted and analyzed considering all available information during the study period. Lower limb spasticity includes the lower limb only. Upper limb spasticity includes the upper limb only. * Some patients exhibited multifocal muscle group involvement, resulting in the total count (*n*) and percentage not aligning with the total number of patients with upper or lower limb spasticity. The denominator of percentages of the subcategories in bold is based on the N of the overall sample—patients treated with BoNT-A and patients not treated with BoNT-A. The denominator of percentages in indented rows is based on the N of the parent category. ^&^ The data reflect the number (*n*) and percentage (%) of patients with available information for this variable.

**Table 3 toxins-16-00340-t003:** Concomitant treatments in patients with PSS.

	Overall PSS (*n* = 2190)	BoNT-A (*n* = 559)	Non-BoNT-A (*n* = 1631)
Non-pharmacological, *n* (%) ^&^	1122 (51.2)	285 (51.0)	837 (51.3)
Rehabilitation and physiotherapy	1072 (95.5)	261 (91.6)	811 (96.9)
Others	69 (6.1)	23 (8.1)	46 (5.5)
Casts	51 (4.5)	17 (6.0)	34 (4.1)
Cryotherapy	43 (3.8)	20 (7.0)	23 (2.7)
Pharmacological, *n* (%) ^&^	798 (36.4)	252 (45.1)	546 (33.5)
Diazepam	363 (45.4)	119 (47.2)	244 (44.7)
Pregabalin	271(34.0)	77 (30.6)	194 (35.5)
Gabapentin	255 (32.0)	88 (34.9)	167 (30.6)
Baclofen (oral or intrathecal)	150 (18.8)	54 (21.4)	96 (17.6)
Others	72 (9.0)	26 (10.3)	46 (8.4)

In some cases, patients had more than one treatment so the sum of patients might add up to more than 100% in each section. The denominator of percentages of the subcategories in bold is based on the N of the overall sample—patients treated with BoNT-A and patients not treated with BoNT-A. The denominator of percentages in indented rows is based on the N of the parent categories. ^&^ The data reflect the number (*n*) and percentage (%) of patients with available information for this variable.

**Table 4 toxins-16-00340-t004:** Performance of EHRead^®^ identifying key variables contained in EHRs.

Variable	Precision	Recall	F1-Score
Intramuscular infiltration of BoNT-A	0.992	0.967	0.979
Hemorrhagic cerebrovascular accident	0.993	0.772	0.869
Spasticity	1.000	0.769	0.870
Equinus foot	0.946	0.779	0.855
Acquired claw toes	1.000	0.667	0.800
Lacunar infarct	1.000	0.632	0.774
Cardioembolic cerebrovascular accident	0.984	0.602	0.747
Thromboembolic cerebrovascular accident	0.867	0.542	0.667
Oral baclofen	1.000	0.406	0.578

## Data Availability

Qualified researchers can request access to summary tables as no individual patient-level data are available. Additional relevant study documents, such as the clinical study report, study protocol with amendments, statistical analysis plan, and dataset specifications, may also be made available. Study documents will be redacted to ensure the privacy of study participants. Requests should be submitted to www.vivli.org (accessed on 31 December 2019) for assessment by an independent scientific review panel. Where applicable, data from eligible studies are available 6 months after the primary manuscript describing the results has been accepted for publication. Further details on Ipsen’s criteria for sharing, eligible studies, and the process for data sharing can be found here (https://vivli.org.members/ourmembers/, accessed on 31 December 2019).

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
