# Peer review of "Botulinum Toxin Type A (BoNT-A) Use for Post-Stroke Spasticity: A Multicenter Study Using Natural Language Processing and Machine Learning"

_toxins, 2024, doi:10.3390/toxins16080340_

Round 1

Reviewer 1 Report

Comments and Suggestions for Authors

please do major revision on the points I have suggested. 

Lack of detailed information on the specific BoNT-A injection sites and dosages used for treatment. Limited information on the specific pain control outcomes and how they were measured. Low percentage of patients with recorded monitoring scales, which may indicate inadequate tracking of treatment progress and effectiveness. Absence of information on potential adverse effects or complications associated with BoNT-A treatment for post-stroke spasticity. Corrections and Recommendations for Improvement: Enhance data documentation in EHRs by including detailed information on BoNT-A injection sites, dosages, and frequency of administration for each patient. Implement standardized pain control outcome measures to accurately assess the effectiveness of BoNT-A treatment for post-stroke spasticity. Increase the use of monitoring scales and ensure consistent tracking of patient progress to evaluate the impact of BoNT-A treatment over time. Conduct further research to investigate and report potential adverse effects or complications associated with BoNT-A use in treating post-stroke spasticity, providing a more comprehensive understanding of the treatment's safety profile.

Comments on the Quality of English Language

N/A

Author Response

Please do major revision on the points I have suggested.

Lack of detailed information on the specific BoNT-A injection sites and dosages used for treatment.
Response: 

Thank you for your valuable feedback. We acknowledge the clinical significance of detailed information about the specific BoNT-A injection sites and dosages used for treatment in our cohort of patients with post-stroke spasticity (PSS). Because of that, those variables were included in the study design and were searched by our EHRead® technology, which can extract and organize free-text information from electronic health records (EHRs) in structured databases. However, this methodology is based on the data jotted down by healthcare workers in the patients’ EHRs, and unfortunately, specific variables such as injection sites and dosages were not usually reported. In this regard, the high percentage of missing values of these variables impeded our ability to derive reliable results. To address this limitation, we have included a sentence in the Discussion section (limitations) that underscores this constraint of our study, and we have improved the writing of the methods section highlighting the unstructured and free-text nature of the information used as a data source.

Limited information on the specific pain control outcomes and how they were measured.

Response: Thank you for your observation. We would like to clarify that the primary aim of this study was to describe demographic, clinical characteristics, and treatment patterns of patients with PSS in a real-world setting rather than evaluate outcomes related to BoNT-A treatment effectiveness. The results presented regarding pain control (Figure 3) are only addressed to describe patient treatment goals, reflecting the indication of BoNT-A in addressing specific complications. Consequently, we have no data regarding pain control outcomes. Moreover, our data related to pain relief was extracted based on the mentions of this clinical concept (including their synonyms, alternative names, acronyms, and abbreviations) in the free text, based on the information in EHR generated during routine clinical practice. In the context of this real-world data study (RWD), methods to measure pain control were not usually reported and could not be extracted, in contrast with prospective studies where prespecified variables can be measured and collected at predefined time points. Additionally, since our technology is based on information in EHRs where the sequence of events may not always be confirmed, we cannot infer causality between a medical event and a specific treatment. To ensure these points are clear, we have added these details in the Discussion section (limitations).

Low percentage of patients with recorded monitoring scales, which may indicate inadequate tracking of treatment progress and effectiveness.

Response: As stated in the Discussion section, the absence of reported clinically relevant variables, such as these clinical scores, poses a challenge in determining whether the missing data are due to clinicians who do not report it in the EHRs or because the information was never entered into the EHRs due to prior treatment in a non-hospital clinical setting. The scarce registration of monitoring scales is an interesting finding previously described in other RWD studies (now cited in the manuscript) These indices are recommended for routine clinical practice and are crucial tools for monitoring. However, not finding these reported indices does not necessarily mean that healthcare providers do not use them; it simply means they are not recording them. This gap in clinical documentation may undermine  accurate disease monitoring and outcomes assessment. Therefore, there is still room for improvement in standardizing reporting in EHRs. This concept is discussed in the revised Discussion section.

Absence of information on potential adverse effects or complications associated with BoNT-A treatment for post-stroke spasticity.

Response: Thank you for your insightful comment. As previously mentioned, the study design was not intended to evaluate effectiveness or safety outcomes. Additionally, due to the inherent limitations of the methodology used for data extraction, it is not possible to infer causality between an adverse event and the specific treatment received. This limitation is reflected in the changes made in the revised Discussion section.

Corrections and Recommendations for Improvement: Enhance data documentation in EHRs by including detailed information on BoNT-A injection sites, dosages, and frequency of administration for each patient. Implement standardized pain control outcome measures to accurately assess the effectiveness of BoNT-A treatment for post-stroke spasticity. Increase the use of monitoring scales and ensure consistent tracking of patient progress to evaluate the impact of BoNT-A treatment over time. Conduct further research to investigate and report potential adverse effects or complications associated with BoNT-A use in treating post-stroke spasticity, providing a more comprehensive understanding of the treatment's safety profile

Response: We appreciate your thoughtful recommendations for improving our study. All your comments have been addressed at some point in the manuscript. However, we would like to clarify that due to the nature of our methodology, we face certain limitations that prevent us from implementing these suggestions at this point.

The present study was conducted through the retrospective extraction of predefined and pre-specified variables from the EHRs of participating hospitals. These variables were extracted with our Natural language Processing (NLP) and machine learning (ML) technology (EHRead®) which focuses on extracting and analyzing the available unstructured information in the EHRs based on routine clinical practice. Therefore, it cannot ensure the standardization or completeness of the data due to these inherent characteristics. Using the described artificial intelligence techniques, those predefined variables along with their clinical context, were refined and modeled to be extracted specifically for the study’s objectives. In this regard, the variable design included specifying those terms related to the variables, their alternative names, synonyms, acronyms, attributes, as well as investigating false negative or false positive terms.

On the one hand, variables related to injection sites, dosage, administration frequency, and monitoring scales were predefined but not identified in the free-text data, resulting in a high percentage of missing values. This issue has been addressed in previous responses within this document and is now discussed in the new version of the manuscript. On the other hand, variables related to effectiveness or safety outcomes were not included in the variable design process due to their lack of alignment with the study’s objectives. Therefore, incorporating new objectives with additional variables at this point would require new ethical committee approval, data extraction, and integration, as well as new processing and analysis of the data. In this context, the authors thank the reviewer for this suggestion and will think about it as a next step to consider to include more recent data. To better understand the NLP and ML process of extracting the clinical information from the EHRs, the methods section has been updated and provides more detailed information. Moreover, a new document with supplemental methods information has been included.

Moderate editing of English language required

Response: Thank you for your valuable feedback. We have conducted an extensive review and thorough editing of the manuscript to enhance the clarity and quality of the English language used. We hope these revisions address your concerns and improve the overall readability of the paper.

Reviewer 2 Report

Comments and Suggestions for Authors

In this manuscript, authors have clearly described the clinical and therapeutic picture of patients with post-stroke spasticity (PSS) in Spain.

Results and discussions sections are good and are carried out in a clear and precise manner.

However, I have some comments and suggestions for authors:

1) I would like authors explain to me why the study only took into consideration data from 1 January 2015 to 31 January 2019; why not consider data up to a later date, closest to 2024 (31 December 2023)?; could there have been problems due to the SARS CoV2 pandemic? If the answer is affirmative, authors describe it in the text and, in any case, in the results section, describe what the data highlighted from 1 January 2020 to 31 December 2023, separating data from the rest of the study (January 1, 2015-January 31, 2019)

2) introduction section: delete the sentence from line 46 to line 48 as we know well what botulinum toxin is and where it comes from

3) I think references number is insufficient. More studies should be included, possibly from the last 5-6 years

4) minor revisions of the English language are required

Comments on the Quality of English Language

Author Response

Reviewer #2

In this manuscript, authors have clearly described the clinical and therapeutic picture of patients with post-stroke spasticity (PSS) in Spain. Results and discussions sections are good and are carried out in a clear and precise manner.

Response: We thank the Reviewer for this comment.

However, I have some comments and suggestions for authors:

1) I would like authors explain to me why the study only took into consideration data from 1 January 2015 to 31 January 2019; why not consider data up to a later date, closest to 2024 (31 December 2023)?; could there have been problems due to the SARS CoV2 pandemic? If the answer is affirmative, authors describe it in the text and, in any case, in the results section, describe what the data highlighted from 1 January 2020 to 31 December 2023, separating data from the rest of the study (January 1, 2015-January 31, 2019)

Response: Thank you for your insightful questions. The present study was conducted through the retrospective extraction of real-world data (RWD) from the Electronic Health Records (EHRs) of participating hospitals. To achieve this, the ethics committees authorized a study period that could extend up to the approval date, which initially was December 2018. However, due to some unforeseen delays mostly related to hospital recruitment (five hospitals) and ethics committee approvals, a study period extension to December 2019 was requested and approved. Further delays related to the acquisition of data from participating hospitals may have been influenced by the pandemic situation, which led to the centralization of resources in patient care areas, thereby minimizing research activities in areas other than COVID-19 infection. After data integration, our methodology involved an extensive process that is required for data extraction using NLP, already necessitating multiple iterations for proper algorithm training, data cleaning, and analysis. Following this, the results needed to be reviewed and validated by numerous experts in the field, which further involved some months for the manuscript preparation to the present moment. At this point, despite the possibility of applying for a new protocol amendment to include data from 2020, the decision was not to include this COVID period based on 1) avoiding bias in clinical practices and 2) not including a new objective that was not initially intended. Therefore, incorporating a new study period today would require new ethical committee approval, data extraction, and integration, as well as new processing and analysis. In this context, the authors thank the reviewer for this suggestion and we will consider it as a next step also including potential new objectives in the field of PSS.

2) introduction section: delete the sentence from line 46 to line 48 as we know well what botulinum toxin is and where it comes from

Response: We appreciated your suggestion and have eliminated the sentence accordingly.

3) I think references number is insufficient. More studies should be included, possibly from the last 5-6 years

Response: Thank you for your feedback. We have conducted a comprehensive review of the literature and have added a considerable number of references to relevant studies on the topic, particularly from the last 5-6 years. We believe these additions significantly strengthen the context and background of our research.

4) minor revisions of the English language are required

Response: Thank you for your valuable feedback. We have conducted an extensive review and thorough editing of the manuscript to enhance the clarity and quality of the English language used. We hope these revisions address your concerns and improve the overall readability of the paper.

Reviewer 3 Report

Comments and Suggestions for Authors

Review of the Article Botulinum Toxin Type A (BoNT-A) Use for Post-Stroke Spasticity: A Multicenter Study Using Natural Language Processing and Machine Learning

General characteristics of the article:

The article presents a multicenter analysis conducted in five Spanish hospitals, examining the use of Botulinum Toxin Type A (BoNT-A) to treat post-stroke spasticity (PSS) using data from electronic health records (EHRs). The study employed EHread® technology, incorporating natural language processing and machine learning, to analyze data from 1,233,929 patients, identifying 2,190 individuals with PSS.

Key findings include a median patient age of 69 years, with 52.1% being male, and 63.2% having suffered an ischemic stroke. Of the PSS patients, 25.5% received BoNT-A treatment, primarily for pain management, typically starting one year post-stroke. Rehabilitation was the most common non-pharmacological treatment, used by 95.5% of the cohort, but only 3.3% had recorded monitoring scales.

The study concludes that early initiation of BoNT-A treatment, improved disease monitoring, and better data documentation in EHRs are essential for optimizing patient outcomes. The use of advanced technologies like natural language processing and machine learning in analyzing medical data is highlighted as a significant methodological strength.

The paper is interesting, written generally correctly and addressing important and timely issues, but it will require minor corrections and additions before proceeding further. I provide detailed comments below.

Major comments:

I'm not sure that hiding the authors' names and affiliations is necessary at this stage especially since you refer to specific clinical centers in the manuscript. Besides, all the data can be seen in the similarity report version of the paper.

The introduction is quite short, please expand and add to the most recent (last 5 years) literature on both stroke statistics and machine learning itself as this aspect was presented very laconically.

The paper contains a reference to only 25 literature sources which is far too small a number for a scientific article.

The purpose of the paper was formulated very generally, please clarify the purpose and emphasize the practical application of the results obtained.

Descriptions of the groups in the results section were presented in great detail but issues related to machine learning were almost completely ignored. Please move the results presented in Table A1 to the main body of the article.

Why, with such a diversity of groups, did you decide to use only the F1 measure and not show the ROC curves with their parameters, accuracy or Matthews correlation coefficient (MCC) values ? They are more accurate in cases of unbalanced groups. Authors can find useful information in the works: https://doi.org/10.1186/s12864-019-6413-7

DOI 10.35784/acs-2022-14  

DOI: 10.1109/IC3INA48034.2019.8949568

I suggest expanding the scopes of analysis and presenting more detailed results with the proposed literature.

I suggest adding information on author contributions as recommended by the journal.

 The work contains numerous editorial errors, missing periods and commas, improper use of punctuation and text preparation. I ask you to thoroughly check your work and adjust it to the requirements of the journal before resubmitting it.

After making appropriate corrections and additions to the content and literature, the work can be further processed.

Comments on the Quality of English Language

Minor editing of English language required

Author Response

Reviewer #3

Review of the Article Botulinum Toxin Type A (BoNT-A) Use for Post-Stroke Spasticity: A Multicenter Study Using Natural Language Processing and Machine Learning

General characteristics of the article:

The article presents a multicenter analysis conducted in five Spanish hospitals, examining the use of Botulinum Toxin Type A (BoNT-A) to treat post-stroke spasticity (PSS) using data from electronic health records (EHRs). The study employed EHread® technology, incorporating natural language processing and machine learning, to analyze data from 1,233,929 patients, identifying 2,190 individuals with PSS.

Key findings include a median patient age of 69 years, with 52.1% being male, and 63.2% having suffered an ischemic stroke. Of the PSS patients, 25.5% received BoNT-A treatment, primarily for pain management, typically starting one year post-stroke. Rehabilitation was the most common non-pharmacological treatment, used by 95.5% of the cohort, but only 3.3% had recorded monitoring scales.

The study concludes that early initiation of BoNT-A treatment, improved disease monitoring, and better data documentation in EHRs are essential for optimizing patient outcomes. The use of advanced technologies like natural language processing and machine learning in analyzing medical data is highlighted as a significant methodological strength.

The paper is interesting, written generally correctly and addressing important and timely issues, but it will require minor corrections and additions before proceeding further. I provide detailed comments below.

Response: We thank the reviewer for these comments.

Major comments:

I'm not sure that hiding the authors' names and affiliations is necessary at this stage especially since you refer to specific clinical centers in the manuscript. Besides, all the data can be seen in the similarity report version of the paper.

Response: Thank you for your observation. We initially submitted the article with the authors' names and affiliations included. The names were likely concealed due to the journal's policy. From our side, we have no issue with making the authors' details visible, provided it complies with the journal's established procedures.

The introduction is quite short, please expand and add to the most recent (last 5 years) literature on both stroke statistics and machine learning itself as this aspect was presented very laconically.

Response: Thank you for your suggestions. We have expanded the introduction to include more comprehensive information on stroke statistics, particularly focusing on European and Spanish populations, as these are the contexts in which our study was conducted. Additionally, we have elaborated on the methodology used for data extraction and analysis, providing a more detailed overview of the natural language processing techniques employed, which intrinsically include machine learning for linguistic models training. Relevant literature from the last years has been incorporated to ensure the introduction reflects the current state of research in both stroke statistics and natural language processing and machine learning.

The paper contains a reference to only 25 literature sources which is far too small a number for a scientific article.

Response: Thank you for your feedback. We have conducted a comprehensive review of the literature and have added a considerable number of references to relevant studies on the topic, particularly from the last 5-6 years. We believe these additions significantly strengthen the context and background of our research.

The purpose of the paper was formulated very generally, please clarify the purpose and emphasize the practical application of the results obtained

Response: Thank you for your insightful comments. We have revised the final part of the introduction and the conclusions to clarify the study's purpose and emphasize the practical application of our results.

Descriptions of the groups in the results section were presented in great detail but issues related to machine learning were almost completely ignored. Please move the results presented in Table A1 to the main body of the article.

Response: Thank you for your suggestion. We have moved the results regarding the performance of EHRead® to the Results section in the main body of the manuscript. Additionally, we have added a comprehensive description of EHRead® technology and an explanation of the external evaluation process and we have included in this new version a supplementary methods section.

Why, with such a diversity of groups, did you decide to use only the F1 measure and not show the ROC curves with their parameters, accuracy or Matthews correlation coefficient (MCC) values? They are more accurate in cases of unbalanced groups. Authors can find useful information in the works:

https://doi.org/10.1186/s12864-019-6413-7

DOI 10.35784/acs-2022-14  

DOI: 10.1109/IC3INA48034.2019.8949568

I suggest expanding the scopes of analysis and presenting more detailed results with the proposed literature.

Response: We appreciate these suggestions and would like to clarify that ML was solely utilized during the data extraction process (EHRead® technology) and not for the analysis reported in the results section. To ensure clarity and transparency, we have conducted an extensive review and reorganization of the methods section, distinguishing between the methodologies used for data extraction and those employed for data analysis, which were based on a descriptive analysis.

Regarding the selection of metrics to evaluate EHRead's performance, we are unable to generate a Receiver Operating Characteristic (ROC) curve because the technology's output is not a probability. As a result, there is no adjustable threshold for determining detections, which makes it impossible to calculate ROC curves.

Additionally, we cannot systematically measure true negatives, which are essential for calculating specificity in ROC analysis. We can only identify true positives (TP), false positives (FP), and false negatives (FN). Therefore, we are limited to calculating precision, recall, and the F1 score.

I suggest adding information on author contributions as recommended by the journal.

Response:  Thank you for the suggestion. Similarly, as with the authors' names and affiliations, we provided this information at the time of manuscript submission. The journal has temporarily removed it, likely due to their policies.

 The work contains numerous editorial errors, missing periods and commas, improper use of punctuation and text preparation. I ask you to thoroughly check your work and adjust it to the requirements of the journal before resubmitting it.

Response: Thank you for your valuable feedback. We have conducted an extensive review and thorough editing of the manuscript to enhance the clarity and quality of the English language used. We hope these revisions address your concerns and improve the overall readability of the paper. However, if some issues related to formatting persist, please consider that using the journal's recommended template could be influencing that.

After making appropriate corrections and additions to the content and literature, the work can be further processed.

Response: We thank the reviewer for the time invested in reviewing our study, their suggestions have contributed to a better result.

Round 2

Reviewer 3 Report

Comments and Suggestions for Authors

The authors made corrections according to my comments. The article can be accepted for publication.

Author Response

We are pleased to know that the changes introduced in the manuscript address all your comments. We appreciate the time you dedicated to reviewing our work. Without a doubt, your contributions have significantly helped improve the final version of the paper.